# polyBERT: a chemical language model to enable fully machine-driven ultrafast polymer informatics

Christopher Kuenneth[1,2] & Rampi Ramprasad [1] ✉

Polymers are a vital part of everyday life. Their chemical universe is so large that it presents unprecedented opportunities as well as significant challenges to identify suitable application-specific candidates. We present a complete end-to-end machine-driven polymer informatics pipeline that can search this space for suitable candidates at unprecedented speed and accuracy. This pipeline includes a polymer chemical fingerprinting capability called polyBERT (inspired by Natural Language Processing concepts), and a multitask learning approach that maps the polyBERT fingerprints to a host of properties. polyBERT is a chemical linguist that treats the chemical structure of polymers as a chemical language. The present approach outstrips the best presently available concepts for polymer property prediction based on handcrafted fingerprint schemes in speed by two orders of magnitude while preserving accuracy, thus making it a strong candidate for deployment in scalable architectures including cloud infrastructures.

Polymers are an integral part of our everyday life and instrumental in the progress of technologies for future innovations[1]. The sheer magnitude and diversity of the polymer chemical space provide opportunities for crafting polymers that accurately match application demands, yet also come with the challenge of efficiently and effectively browsing this gigantic space. The nascent field of polymer informatics[2–5] allows access to the depth of the polymer universe and demonstrates the potency of machine learning (ML) models to overcome this challenge. ML frameworks have enabled substantial progress in the development of polymer property predictors[6–10] and solving inverse problems in which polymers that meet specific property requirements are either identified from candidate sets[11,12], or are freshly designed using genetic[13,14] or generative[15–17] algorithms.

An essential step in polymer informatics pipelines is the conversion of polymer chemical structures to numerical representations that are often called fingerprints, features, or descriptors (see blue boxes in Fig. 1a). Past handcrafted fingerprinting approaches[18–22] utilize cheminformatics tools that numerically encode key chemical and structural features of polymers. Although such handcrafted fingerprints build on invaluable intuition and experience, they are tedious to

develop, involve complex computations that often consume most of the time during model training and inference, and lack generalization to all polymer chemical classes (i.e., new features may have to be added to the catalog of features in an ad hoc manner). ML pipelines that use handcrafted fingerprints are thus prone to errors during the exploration of new polymer chemical classes. Also, handcrafted fingerprints present barriers for the development and deployment of fully machine-driven ipipelines, which are suited for scalability in cloud computing and high-throughput environments.

One way to overcome the previously mentioned limitations is to replace handcrafted fingerprints with fully machine-crafted "Transformer" fingerprints (see right pipeline of Fig. 1a). Transformers[23] were recently developed in the field of Natural Language Processing (NLP) and have swiftly become the gold standard in ML language modeling. In this work, we envision Simplified Molecular-Input Line-Entry System (SMILES)[24] strings that have been used to represent polymers as the "chemical language" of polymers. We use millions of Polymer SMILES (PSMILES) strings for training a language model called polyBERT to become an expert—a linguist—of the polymer chemical language. In combination with multitask deep neural networks[6,7], polyBERT enables

[1]School of Materials Science and Engineering, Georgia Institute of Technology, Atlanta, GA 30332, USA. [2]Faculty of Engineering Science, University of Bayreuth, 95447 Bayreuth, Germany. ✉e-mail: rampi.ramprasad@mse.gatech.edu

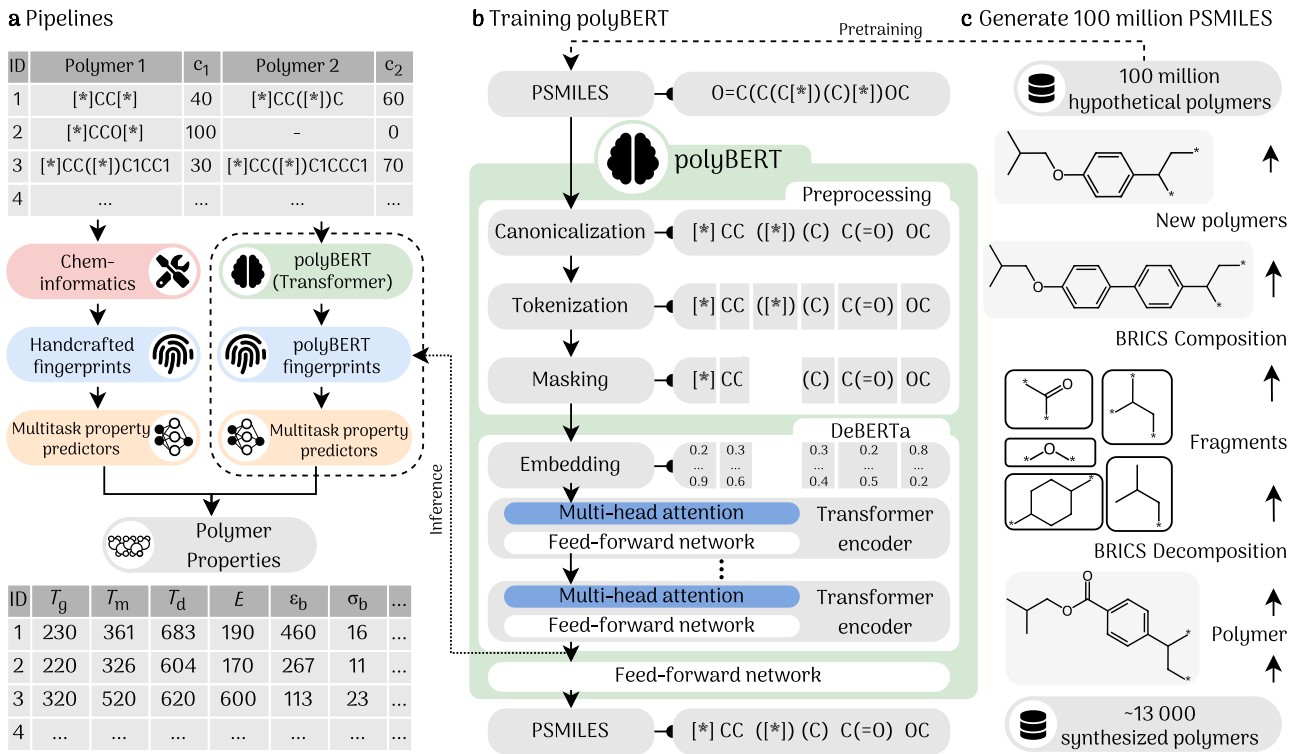

**Fig. 1 | Polymer informatics with polyBERT. a** Prediction pipelines. The left pipeline shows the prediction using handcrafted fingerprints using cheminformatics tools, while the right pipeline (present work) portrays a fully end-to-end machine-driven predictor using polyBERT. Property symbols are defined in Table 1. ID1 and ID3 are copolymers, and ID2 is a homopolymer. $c_1$ and $c_2$ are the fractions of the first and second comonomer in the polymer. The symbols $T_g$, $T_m$, $T_d$, $E$, $\epsilon_b$, and $\sigma_b$ stand for glass transition temperature, melting temperature, degradation temperature, Young's modulus, elongation at break, and tensile strength at break, respectively. **b** polyBERT is a polymer chemical language linguist. polyBERT canonicalizes, tokenizes, and masks Polymer Simplified Molecular-Input Line-Entry System (PSMILES) strings before passing them to the DeBERTa model. Each of the 12 Transformer encoders has 12 attention heads. A last dense layer with a softmax activation function finds the masked tokens. polyBERT fingerprints (dashed arrow) are the averages over the token dimension (sentence average) of the last Transformer encoder. **c** 100 million hypothetical PSMILES strings. First, 13 766 known (i.e., previously synthesized) polymers are decomposed to 4424 fragments using the Breaking Retrosynthetically Interesting Chemical Substructures (BRICS)[40] method. Second, re-assembling the BRICS fragments in many different ways generates 100 million hypothetical polymers by randomly and enumeratively combining the fragments.

a fully end-to-end machine-driven polymer informatics pipeline that uses and unleashes the true power of artificial intelligence methods. Multitask deep neural networks harness inherent correlations in multi-fidelity and multi-property data sets, scale effortlessly in cloud computing environments, and generalize to multiple prediction tasks.

Recent studies[25–27] demonstrated the benefits of using Transformers in the molecule chemical space. For example, Wang et al.[26] have trained a BERT[28] model (the most common general language model) with a data set of molecule SMILES strings. Using BERT's latent space representations of molecules as fingerprints, the authors show that their approach outperforms other fingerprinting methods (including fingerprints of an unsupervised recurrent neural network and a graph neural network). Similarly, Schwaller et al.[29,30] have developed a Transformer model to predict retrosynthesis pathways of molecules from reactants and reagents that outperforms known algorithms in the reaction prediction literature. A very recent study by Xu et al.[31] (performed almost at the same time as us, as can be confirmed by both our arXiv submissions) used a RoBERTa[32] model (an evolution of the BERT Transformer model) for polymer property predictions. Their training strategy first involves the pretraining (unsupervised training) of the RoBERTa[32] model using 5 million polymers and then a finetuning step (supervised training) to directly predict polymer properties. Although their work uses much smaller datasets than ours both for unsupervised and supervised training tasks, they find that their finetuned RoBERTa model outperforms graph neural networks, long short-term memory and other models; we do note that this recent work does not make direct comparisons of their Transformer-based model with the current

state-of-the-art in hand-crafted fingerprinting and multi-task learning[7] (which we do in the present contribution).

Another promising neural network architecture, namely, graph neural networks[33], which treats chemical structures as graphs, has been applied to the molecule and polymer chemical space in the past. In contrast to Transformers, graph neural networks represent atoms as nodes and bonds as edges of a graph, so as to encode immediate and extended connectivities between atoms. As a consequence, graph neural networks are not directly based on PSMILES strings like Transformers, but depend on an initial set of feature vectors (such as atom types, implicit valence, etc.) that need to be computed for and assigned to each node. For example, Park et al.[34] compared predictions of a graph convolutional network and the popular extended-connectivity circular fingerprint[19] for thermal and mechanical polymer properties, finding a similar prediction performance for both models. Similarly, Gurnani et al.[35] used multitask graph neural networks to predict polymer properties, but introduced edges between the heavy boundary atoms to incorporate the recurrent topology of polymer chains. Their combined approach of graph neural networks and multitask learning outperforms predictions based on the conventional handcrafted Polymer Genome fingerprint[8,20] in almost all cases. In a similar manner, Aldeghi and Coley[36] introduced low-weight edges between polymer chains to enable predictions for alternating, random, and block copolymers, and termini chemical groups. We also note that unlike Transformers graph neural networks are usually trained end-to-end, i.e., their latent space representations (fingerprints) are learned under supervision with polymer properties. The

consequence of this is that in the case of Transformer-based approaches, the learned fingerprint is independent of the polymer properties (and so can be determined once and for all), where as graph neural network architectures are typically constructed such that the learned representations depend on the specific property under consideration. We note that self-supervised graph neural networks[37,38] have recently been developed that learn the molecule graph through atom, bond, and subgraph masking, an approach similar to Transformers.

This work has several critical ingredients. First, we generate a data set of 100 million hypothetical polymers by enumeratively combining chemical fragments extracted from a list of more than 13 000 synthesized polymers. Next, we train polyBERT, a DeBERTa[39]-based encoder-only Transformer, using this hypothetical polymer data set to become a polymer chemical linguist. During training, polyBERT learns to translate input PSMILES strings to numerical representations that we use as polymer fingerprints. Finally, we map the polyBERT fingerprints to about 3 dozen polymer properties using our multitask ML framework to yield fully machine-driven ultrafast polymer property predictors. For benchmarking, the performance (both accuracy and speed) of this new end-to-end property prediction pipeline is compared with the state-of-the-art handcrafted Polymer Genome[8] (PG) fingerprint based pipeline pioneered previously. Using the ultrafast polyBERT polymer informatics pipeline, we are in a position to predict the properties of the 100 million hypothetical polymers intending to find property boundaries of the polymer universe. This work contributes to expediting the discovery, design, development, and deployment of polymers by harnessing the true power of language, data, and artificial intelligence models.

## Results

### Data sets

Figure 1c sketches the two-step process for fabricating 100 million hypothetical PSMILES strings. We use the Breaking Retrosynthetically Interesting Chemical Substructures (BRICS)[40] method (as implemented in RDKit[41]) to decompose previously synthesized 13,766 polymers (all monomers of the data set outline in Table 1, see below) into 4424 unique chemical fragments. Random and enumerative compositions of these fragments yield 100 million hypothetical PSMILES strings that we first canonicalize (see "Methods" section) and then use for training polyBERT. The hypothetical PSMILES strings are chemically valid polymers but, mostly, have never been synthesized before.

Once polyBERT has completed its unsupervised learning task using the 100 million hypothetical PSMILES strings, multitask supervised learning maps polyBERT polymer fingerprints to multiple properties to produce property predictors. We use the property data set in Table 1 for training the property predictors. The data set contains 28,061 (≈80%) homopolymer and 7456 (≈20%) copolymer (total of 35,517) data points of 29 experimental and computational polymer properties that pertain to 11,145 different monomers and 1338 distinct copolymer chemistries, respectively. Each of the 7456 copolymer data points involves two distinct comonomers at various compositions. Our copolymer data points are for random copolymers, which are adequately handled by our adopted fingerprinting strategy (see "Methods" section). Alternating copolymers are treated as homopolymers with appropriately defined repeat units for fingerprinting purposes. Other flavors of copolymers may also be encoded by adding additional fingerprint components. All data points in the data set have been used in past studies[6,7,11,42–49] and were produced using computational methods or obtained from literature and other public sources. Supplementary Figs. S3–S8 show histograms for each property.

### polyBERT

polyBERT iteratively ingests 100 million hypothetical PSMILES strings to learn the polymer chemical language, as sketched in Fig. 1b. Using

100 million PSMILES strings is the latest example of training a chemistry-related language model with a large data set and follows the trend of growing data sets in this discipline, with ChemBERTa using 10 million, SMILES-BERT using 18.7 million, and ChemBERTa-2 using 77 million SMILES strings.[50] polyBERT is a DeBERTa[39] model (as implemented in Huggingface's Transformer Python library[51]) with a supplementary three-stage preprocessing unit for PSMILES strings. We chose the DeBERTa model as the foundation of polyBERT because it outperformed other BERT-like models (BERT[28], RoBERTa[32], and DistilBERT[52]) in our tests (see Supplementary Discussion) and standardized performance task[39]. First, polyBERT transforms a input PSMILES string into its canonical form (e.g., `[*]CCOCCO[*]` to `[*]COC[*]`) using the `canonicalize_psmiles` Python package developed in this work. Details can be found in the Methods section. Second, polyBERT tokenizes canonical PSMILES strings using the SentencePiece[53] tokenizer and a total of 265 tokens. The tokens include common PSMILES characters such as the uppercased and lowercased 118 elements of the periodic table of elements, numbers ranging from 0 to 9, and special characters like `[*]`, `(`, `)`, `=`, among others. This ensures that the tokenizer covers the entire PSMILES strings vocabulary and is a similar approach to that in ref. 50. A full token list can be found at the GitHub repository (see the Data and Code Availability section). Third, polyBERT masks 15% (default parameter for masked language models) of the tokens to create a self-supervised training task. In this training task, polyBERT is taught to predict the masked tokens using the non-masked surrounding tokens by adjusting the weights of the Transformer encoders (fill-in-the-blanks task). We use 80 million PSMILES strings for training and 20 million PSMILES strings for validation. The validation F1-score is > 0.99. This exceptionally good F1-score indicates that polyBERT finds the masked tokens in almost all cases. The total $CO_2$ emissions for training polyBERT on our hardware are estimated to be 12.6 kg$CO_2$eq (see $CO_2$ Emission and Timing section).

The training with 80 million PSMILES strings renders polyBERT an expert polymer chemical linguist who knows grammatical and syntactical rules of the polymer chemical language. polyBERT learns patterns and relations of tokens via the multi-head self-attention mechanism and fully connected feed-forward network of the Transformer encoders[23]. The attention mechanism instructs polyBERT to devote more focus to a small but essential part of a PSMILES string. polyBERT's learned latent spaces after each encoder block are numerical representations of the input PSMILES strings. The polyBERT fingerprint is the average over the token dimension (sentence average) of the last latent space (dotted line in Fig. 1b). We use the Python package SentenceTransformers[54] for extracting and computing polyBERT fingerprints.

### Fingerprints

For acquiring analogies and juxtaposing chemical relevancy, we compare polyBERT fingerprints with the handcrafted Polymer Genome[8] (PG) fingerprints that numerically encode polymers at three different length scales. A description of PG fingerprints can be found in "Methods" section. The PG fingerprint vector for the data set in this work has 945 components and is sparsely populated (93.9% zeros). The reason for this ultra sparsity is that many PG fingerprint components count chemical groups in polymers[8]. A fingerprint component of zero indicates that a chemical group is not present. In contrast, polyBERT fingerprint vectors have 600 components and are fully dense (0% zeros). Fully dense and lower-dimensional fingerprints are often advantageous for ML models whose computation time scales superlinear ($\mathcal{O}(n^s)$, $s > 1$) with the data set size ($n$) such as Gaussian process or kernel ridge techniques. Moreover, in the case of neural networks, sparse and high-dimensional input vectors can cause unnecessary high memory load that reduces training and inference speed. We note that the dimensionality of polyBERT fingerprints is a parameter that can be

**Table 1 | Training data set for the property predictors. The properties are sorted into categories, showed at the top of each block. The data set contains 29 properties (dielectric constants $k_f$ are available at 9 different frequencies $f$). HP and CP stand for homopolymer and copolymer, respectively**

| Property | Symbol | Unit | Source[a] | Data range | Data points | | |
|---|---|---|---|---|---|---|---|
| | | | | | HP | CP | All |
| **Thermal** | | | | | | | |
| Glass transition temp. | $T_g$ | K | Exp. | [8e+01, 9e+02] | 5183 | 3312 | 8495 |
| Melting temp. | $T_m$ | K | Exp. | [2e+02, 9e+02] | 2132 | 1523 | 3655 |
| Degradation temp. | $T_d$ | K | Exp. | [3e+02, 1e+03] | 3584 | 1064 | 4648 |
| **Thermodynamic & physical** | | | | | | | |
| Heat capacity | $c_p$ | $Jg^{-1}K^{-1}$ | Exp. | [8e-01, 2e+00] | 79 | | 79 |
| Atomization energy | $E_{at}$ | eV atom$^{-1}$ | DFT | [-7e+00, -5e+00] | 390 | | 390 |
| Limiting oxygen index | $O_i$ | % | Exp. | [1e+01, 7e+01] | 101 | | 101 |
| Crystallization tendency (DFT) | $X_c$ | % | DFT | [1e-01, 1e+02] | 432 | | 432 |
| Crystallization tendency (exp.) | $X_e$ | % | Exp. | [1e+00, 1e+02] | 111 | | 111 |
| Density | $\rho$ | g cm$^{-3}$ | Exp. | [8e-01, 2e+00] | 910 | | 910 |
| **Electronic** | | | | | | | |
| Band gap (chain) | $E_{gc}$ | eV | DFT | [2e-02, 1e+01] | 4224 | | 4224 |
| Band gap (bulk) | $E_{gb}$ | eV | DFT | [4e-01, 1e+01] | 597 | | 597 |
| Electron affinity | $E_{ea}$ | eV | DFT | [4e-01, 5e+00] | 368 | | 368 |
| Ionization energy | $E_i$ | eV | DFT | [4e+00, 1e+01] | 370 | | 370 |
| Electronic injection barrier | $E_{ib}$ | eV | DFT | [2e+00, 7e+00] | 2610 | | 2610 |
| Cohesive energy density | $\delta$ | cal cm$^{-3}$ | Exp. | [2e+01, 3e+02] | 294 | | 294 |
| **Optical & dielectric** | | | | | | | |
| Refractive index (DFT) | $n_c$ | | DFT | [1e+00, 3e+00] | 382 | | 382 |
| Refractive index (exp.) | $n_e$ | | Exp. | [1e+00, 2e+00] | 516 | | 516 |
| Dielec. constant (DFT) | $k_c$ | | DFT | [3e+00, 9e+00] | 382 | | 382 |
| Dielec. constant at freq. **f**[b] | $k_f$ | | Exp. | [2e+00, 1e+01] | 1187 | | 1187 |
| **Mechanical** | | | | | | | |
| Young's modulus | $E$ | MPa | Exp. | [2e-02, 4e+03] | 592 | 322 | 914 |
| Tensile strength at yield | $\sigma_y$ | MPa | Exp. | [3e-05, 1e+02] | 216 | 78 | 294 |
| Tensile strength at break | $\sigma_b$ | MPa | Exp. | [5e-03, 2e+02] | 663 | 318 | 981 |
| Elongation at break | $\epsilon_b$ | | Exp. | [3e-01, 1e+03] | 868 | 260 | 1128 |
| **Permeability** | | | | | | | |
| $O_2$ gas permeability | $\mu_{O_2}$ | barrer | Exp. | [5e-06, 1e+03] | 390 | 210 | 600 |
| $CO_2$ gas permeability | $\mu_{CO_2}$ | barrer | Exp. | [1e-06, 5e+03] | 286 | 119 | 405 |
| $N_2$ gas permeability | $\mu_{N_2}$ | barrer | Exp. | [3e-05, 5e+02] | 384 | 99 | 483 |
| $H_2$ gas permeability | $\mu_{H_2}$ | barrer | Exp. | [2e-02, 5e+03] | 240 | 46 | 286 |
| He gas permeability | $\mu_{He}$ | barrer | Exp. | [5e-02, 2e+03] | 239 | 58 | 297 |
| $CH_4$ gas permeability | $\mu_{CH_4}$ | barrer | Exp. | [4e-04, 2e+03] | 331 | 47 | 378 |
| | | | | | 28,061 | 7456 | 35,517 |

[a]Experiments (Exp.); density functional theory (DFT).
[b]**f** $\in$ {1.78,2,3,4,5,6,7,9,15} is the $\log_{10}$(frequency in Hz); e.g., $k_3$ is the dielectric constant at a frequency of 1 kHz.

chosen arbitrarily to yield the best training result. A summary of the key figures can be found in Supplementary Table S4.

Figure 2 shows Uniform Manifold Approximation and Projection (UMAP)[55] plots for all homo- and copolymer chemistries in Table 1. The colored triangles in the first column indicate the coordinates of three selected polymers for polyBERT and PG fingerprints. We observe for both fingerprint types that the orange and blue triangles are very close, while the green triangle is separate. We also note that polymers corresponding to the orange and blue triangles, namely poly(but-1-ene) and poly(pent-1-ene), have similar chemistry (different by only one carbon atom), but poly(4-vinylpyridine) represented by a green triangle, is different. This chemically intuitive positioning of fingerprints suggests the chemical relevancy of fingerprint distances. The cosine fingerprint distances reported in Supplementary Fig. S1 allow for the same conclusion.

The second, third, and fourth columns of Fig. 2 display the same UMAP plots as in the first column. Colored dots indicate the property values of $T_g$, $T_d$, and $E_{gc}$, while light gray dots show polymer fingerprints with unknown property values. We observe localized clusters of similar color in each plot pertaining to polymers of similar properties. Although this finding is not surprising for the PG fingerprint because it relies on handcrafted chemical features that purposely position similar polymers next to each other, it is remarkable for polyBERT. With no chemical information and purely based on training on a massive amount of PSMILES strings, polyBERT has learned polymer fingerprints that match chemical intuition. This again shows that polyBERT fingerprints have chemical pertinence and their distances measure polymer similarity (e.g., using the cosine distance metric).

polyBERT learns chemical motifs and relations in the PSMILES strings using the Transformer encoders, each of which includes an

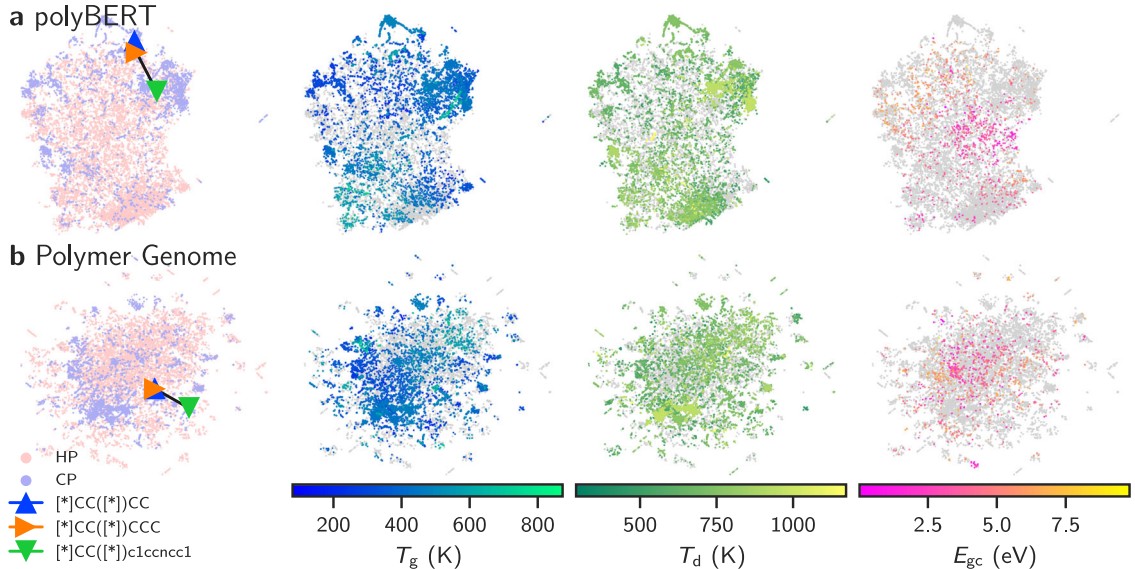

**Fig. 2 | Two-dimensional Uniform Manifold Approximation and Projection[55] (UMAP) plots of the fingerprints.** Panel **a** shows polyBERT and panel **b** shows Polymer Genome fingerprints for all homo- and copolymer chemistries in Table 1. The triangles (blue, orange, and green) in the first column indicate fingerprint positions in the UMAP spaces of three selected polymers. The colored dots in columns two, three, and four indicate property values of $T_g$, $T_d$, and $E_{gc}$, which stand for the glass transition temperature, degradation temperature, and band gap (chain), respectively. Light gray dots show polymers with unknown property values. The Polymer Simplified Molecular-Input Line-Entry System (PSMILES) strings `[*]CC([*])CC`, `[*]CC([*])CCC`, and `[*]CC([*])c1ccncc1` denote poly(but-1-ene), poly(pent-1-ene), and poly(4-vinylpyridine), respectively.

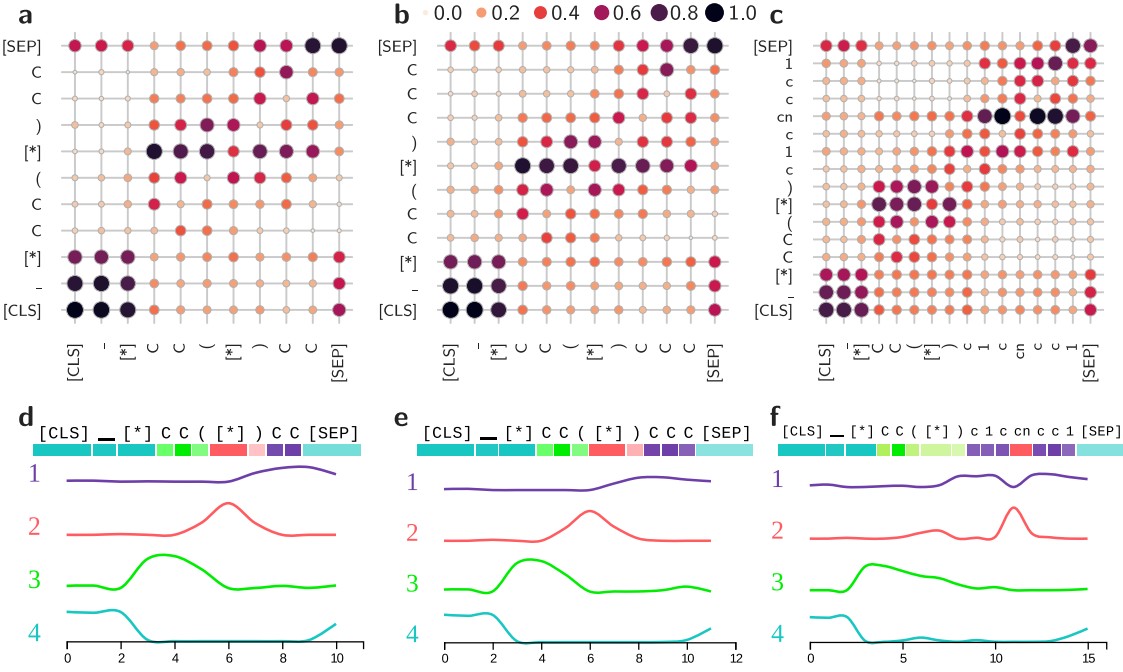

**Fig. 3 | Attention maps and neuron activation for three polymers.** Panels **a**–**c** show the normalized attention maps summed over all 12 attention heads and 12 encoders of polyBERT. Panels **d**–**f** show the factorized neurons activations in the feed-forward network layers[56]. The Polymer Simplified Molecular-Input Line-Entry System (PSMILES) strings `[*]CC([*])CC`, `[*]CC([*])CCC`, and `[*]CC([*])c1ccncc1` denote poly(but-1-ene), poly(pent-1-ene), and poly(4-vinylpyridine), respectively.

attention and feed-forward network layer (see Fig. 1b). Figure 3a–c displays the normalized attention maps summed over all 12 attention heads and 12 encoders of polyBERT for the same PSMILES strings as in Fig. 2. Large dots indicate high attention scores, while small dots show weak attention scores. The attention scores can be interpreted as the importance of knowing the position and type of another token (or chemical motif) and its impact on the current token's latent space. The [CLS], _, and [SEP] tokens are auxiliary tokens. The first two tokens indicate the beginning of PSMILES strings and the last token shows the end of PSMILES strings. We notice high attention scores for the [CLS], _, and first [*] tokens in all panels a to c that imply the connection of the auxiliary tokens to the beginning of PSMILES strings. Also, we observe at least intermediate attention scores to next and next-to-next neighbors (first and second off-

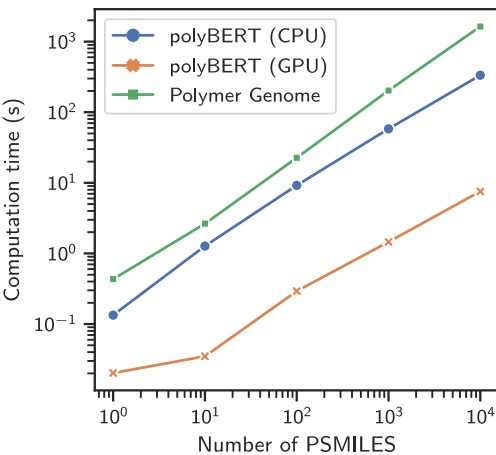

**Fig. 4 | Computation times of polymer fingerprints.** The fingerprints are computed on one CPU core (Intel(R) Xeon(R) CPU E5-2667), except for polyBERT (GPU) fingerprints that are computed on one GPU (Quadro GP100). The computation times per Polymer Simplified Molecular-Input Line-Entry System (PSMILES) string, in the order of the legend, are 33.39, 0.76, and 163.59 ms/PSMILES (computed for $10^4$ PSMILES), respectively.

diagonal elements) for all tokens highlighting the importance of closely bonded neighbors for the polyBERT fingerprint. Another general trend is large attention scores between the second `[*]` tokens and multiple neighbor tokens across all panels. Moreover, in Fig. 3c, we find large attention scores for the `cn` token up to the fourth or fifth neighbor tokens that indicate a strong impact of `cn` to the latent spaces and polyBERT fingerprint, which is expected due to the different nature of the nitrogen atom.

Figure 3d–f shows the non-negative matrix factorizations (4 components) of the neuron activations in the feed-forward neural network layers[56] of polyBERT (see Fig. 1b) for the same polymers as in panels a to c. The neurons in the feed-forward network layers account for more than 60% of the parameters. Each of the four components represent a set of distinct neurons that is active for specific tokens (x-axes). For example, the fourth set of neurons is active if polyBERT predicts latent spaces for the auxiliary tokens. The third set of neurons fire in the case of the first two `C` tokens and the first set of neurons are active for side chain `c` or `C` atoms, except in the case of the `cn` token, which has its own set of neurons (second set of neurons). In total, the attention layers incorporate positional and relational knowledge and the feed-forward neural network layers disable and enable certain routes through polyBERT. Both factors modulate the polyBERT fingerprints.

Not surprisingly, the computations of polyBERT and PG fingerprints scale nearly linearly with the number of PSMILES strings although their performance (i.e., pre-factor) can be quite different, as shown in the log-log scaled Fig. 4. The computation of polyBERT (GPU) is over two orders of magnitude (215 times) faster than computing PG fingerprints. polyBERT fingerprints may be computed on CPUs and GPUs. Because of the presently large efforts in industry to develop faster and better GPUs, we expect the computation of polyBERT fingerprint to become even faster in the future. Time is very important for high-throughput polymer informatics pipelines that identify polymers from large candidate sets[11]. With an estimate of 0.30 ms/PSMILES for the multitask deep neural networks (see Property Prediction section), the total time using the polyBERT-based pipeline to predict 29 polymer properties sums to 1.06 ms/polymer/GPU.

## Property prediction
For benchmarking the property prediction accuracy of polyBERT and PG fingerprints, we train multitask deep neural networks for each

property category defined in Table 1. In our previous study[7], we observed that these property categories resulted in the development of models exhibiting superior performance. Multitask deep neural networks have demonstrated best-in-class results for polymer property predictions[6,7,11], while being fast, scalable, and readily amenable if more data points become available. Unlike single-task models, multitask models simultaneously predict numerous properties (tasks) and harness inherent but hidden correlations in data to improve their performance. Such correlation exists, for instance, between $T_g$ and $T_m$, but the exact correlation varies across specific polymer chemistries. Multitask models learn and improve from these varying correlations in data. The training protocol of the multitask deep neural networks follows state-of-the-art methods involving five-fold cross-validation and a consolidating meta learner that forecasts the final property values based upon the ensemble of cross-validation predictors. More details about multitask deep neural networks are provided in the Methods section. Their training process is outlined in Supplementary Figure S2.

Figure 5a shows the coefficient of determination ($R^2$) averages and standard deviations across the five validation data sets of the cross-validation process for 29 polymer properties. The averages are independent of the data set splits, while the standard deviations show the variance of the prediction performance for the different splits. Smaller standard deviations indicate data sets with homogeneously distributed data points in the learning space. Large standard deviations stem from inhomogeneously distributed data points of usually smaller data sets. Cross-validation is shown to establish an independence of the data set splits for polymer predictions[11]. Root-Mean-Square Error (RMSE) and $R^2$ values for the cross-validation and meta learner models can be found in Supplementary Table S1–S3 for all polymers, homopolymers, and copolymers, respectively. We find the prediction accuracy to be better for thermal and mechanical properties of copolymers (relative to that for homopolymers) and slightly worse for the gas permeabilities, similar to previous findings[6]. Overall, PG performs best ($R^2 = 0.81$) but is very closely followed by polyBERT ($R^2 = 0.80$). This overall performance order of the fingerprint types is persistent with the category averages and properties, except for $X_c$, $X_e$, and $\epsilon_b$, where polyBERT slightly outperforms PG fingerprints. We note that polyBERT and PG fingerprints are both practical routes for polymer featurization because their $R^2$ values lie close together and are generally high. polyBERT fingerprints have the accuracy of the handcrafted PG fingerprints but are over two orders of magnitude faster (see Fig. 4).

Figure 5b shows high $R^2$ values for each meta learner (one for each category), suggesting an exceptional prediction performance across all properties. We train the meta learners on unseen 20% of the data set and validate using 80% of the data set (also used for cross-validation). The reported validation $R^2$ values thus only partly measure the generalization performance with respect to the full data set. Meta learners can be conceived as taking decisive roles in selecting the best values from the predictions of the five cross-validation models. We use the meta learners for all property predictions in this work. Supplementary Figs. S9–S14 show the meta learners' parity plots.

The ultrafast and accurate polyBERT-based polymer informatics pipeline allows us to predict all 29 properties of the 100 million hypothetical polymers that were originally created to train polyBERT. Figure 5c shows the minimum, mean, and maximum for each property. Histograms are given in Supplementary Figs. S15–S20. Given the vast size of our data set and consequent chemical space of the 100 million hypothetical polymers, the minimum and maximum values can be interpreted as potential boundaries of the total polymer property space. In addition, a data set of this magnitude presents numerous opportunities for obtaining fascinating insights and practical applications. For example, it can be utilized in future studies to establish standardized benchmarks for testing and evaluating ML models in the domain of polymer informatics. The data set may also reveal structure-property information that provides guidance for design rules, helps to

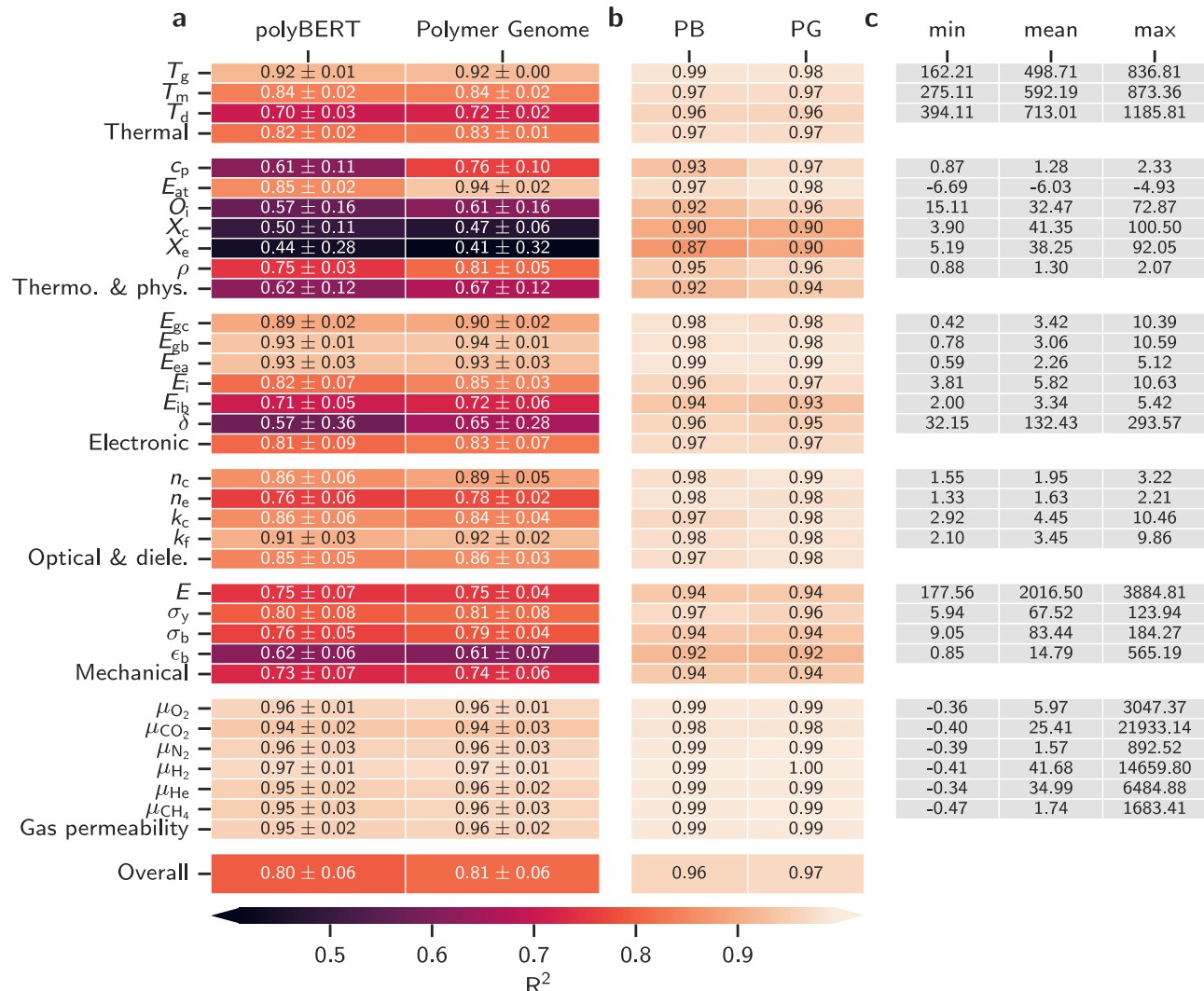

**Fig. 5 | Coefficient of determination ($R^2$) performance values for polyBERT (PB) and Polymer Genome (PG) fingerprints.** Panel **a** shows $R^2$ averages of the five cross-validation validation data sets along with standard deviations ($1\sigma$) and panel **b** shows $R^2$ values of the meta learner's test data set. The category-averaged $R^2$ values are stated in the last rows of each block, while overall $R^2$ values are given in the very last block. The properties gas permeabilities ($\mu_x$) and elongation at break ($\epsilon_b$) are trained on log base 10 scale ($x \mapsto \log_{10}(x+1)$). The $R^2$ values are reported on this scale. Panel **c** shows the minimum, mean, and maximum of polyBERT-based property predictions for 100 million hypothetical polymers. $T_g$, $T_m$, and $T_d$ stand for glass transition, melting, and degradation temperature. $c_p$, $E_{at}$, $O_i$, $X_c$, $X_e$, and $\rho$

stand for heat capacity, atomization energy, limiting oxygen index, crystallization tendency (DFT), crystallization tendency (exp.), and density. $E_{gc}$, $E_{gb}$, $E_{ea}$, $E_i$, $E_{ib}$, and $\delta$ stand for band gap (chain), band gap (bulk), electron affinity, ionization energy, electronic injection barrier, and cohesive energy density. $n_c$, $n_e$, $k_c$, and $k_f$ stand for refractive index (DFT), refractive index (exp.), dielectric constant (DFT), and dielectric constant at freq. $f \in \{1.78,2,3,4,5,6,7,9,15\}$. $E$, $\sigma_y$, $\sigma_b$, and $\epsilon_b$ stand for Young's modulus, tensile strength at yield, tensile strength at break, and elongation at break. $\mu_{O_2}$, $\mu_{CO_2}$, $\mu_{N_2}$, $\mu_{H_2}$, $\mu_{He}$, and $\mu_{CH_4}$ stand for $O_2$, $CO_2$, $N_2$, $H_2$, He, and $CH_4$ gas permeability. Plain numbers of this Figure can be found in Supplementary Tables S1 and S5.

identify unexplored areas to search for new polymers, or facilitates direct selection of polymers with specific properties through nearest neighbor searches, as evidenced in a recent study[11]. A possible future evolution of the data set may also contain subspaces of distinct polymer classes, such as biodegradable or low-carbon polymer classes. However, these aspects are beyond the scope of this study. The data set with 100 million hypothetical polymers including the predictions of 29 properties is available for academic use. The total $CO_2$ emissions for predicting 29 properties of 100 million hypothetical polymers are estimated to be 5.5 kg$CO_2$eq (see $CO_2$ Emission and Timing section).

**Other advantages of polyBERT: beyond speed and accuracy**
The feed-forward network (last layer in Fig. 1b), which predicts masked tokens during the self-supervised training of polyBERT, enables the mapping of numerical latent spaces (i.e., fingerprints) to PSMILES strings. However, because we average over the token dimension of the

last latent space to compute polyBERT fingerprints, we cannot unambiguously map the current fingerprints back to PSMILES strings. A modified future version of polyBERT that provides PSMILES strings encoding and fingerprint decoding could involve inserting a dimensionality-reducing layer after the last Transformer encoder. Fingerprint decoders are important elements of design informatics pipelines that invert the prediction pipeline to meet property specifications. We note that the current choice of computing polyBERT fingerprints as pooling averages stems from basic dimensionality reduction considerations that require no modification of the DeBERTa architecture.

A second advantage of the polyBERT approach is interpretability. Analyzing the chemical relevancy of polyBERT fingerprints (as discussed in the Fingerprints section) in greater detail can reveal chemical functions and interactions of structural parts of the polymers. As illustrated with the examples of the three polymers in Fig. 3, deciphering

and visualizing the attention layers of the Transformer encoders can reveal such information. Saliency methods[57] may also be used to directly explain the relationships between structural parts of the PSMILES strings (inputs) and polymer properties (outputs).

Yet another advantage of the polyBERT approach is its coverage of the entire chemical space. Molecule SMILES strings are a subset of polymer SMILES strings and differ by only two stars (`[*]`) symbols that indicate the two endpoints of the polymer repeat unit. polyBERT has no intrinsic limitations or functions that obstruct predicting fingerprints for molecule SMILES strings. Our first experiments show consistent and well-conditioned fingerprints for molecule SMILES strings using polyBERT that required only minimal changes in the canonicalization routine.

## Discussion

Here, we show a generalizable, ultrafast, and accurate polymer informatics pipeline that is seamlessly scalable on cloud hardware and suitable for high-throughput screening of huge polymer spaces. polyBERT, which is a Transformer-based NLP model modified for the polymer chemical language, is the critical element of our pipeline. After training on 100 million hypothetical polymers, the polyBERT-based informatics pipeline arrives at a representation of polymers and predicts polymer properties over two orders of magnitude faster but at the same accuracy as the best pipeline based on handcrafted PG fingerprints.

The total polymer universe is gigantic, but currently limited by experimentation, manufacturing techniques, resources, and economical aspects. Contemplating different polymer types such as homopolymers, copolymer, and polymer blends, novel undiscovered polymer chemistries, additives, and processing conditions, the number of possible polymers in the polymer universe is truly limitless. Searching this extraordinarily large space enabled by property predictions is limited by the prediction speed. The accurate prediction of 29 properties for 100 million hypothetical polymers in a reasonable time demonstrates that polyBERT is an enabler to extensive explorations of this gigantic polymer universe at scale. polyBERT paves the pathway for the discovery of novel polymers 100 times faster (and potentially even faster with newer GPU generations) than state-of-the-art informatics approaches – but at the same accuracy as slower handcrafted fingerprinting methods—by leveraging Transformer-based ML models originally developed for NLP. polyBERT fingerprints are dense and chemically pertinent numerical representations of polymers that adequately measure polymer similarity. They can be used for any polymer informatics task that requires numerical representations of polymers such as property predictions (demonstrated here), polymer structure predictions, ML-based synthesis assistants, etc. polyBERT fingerprints have a huge potential to accelerate past polymer informatics pipelines by replacing the handcrafted fingerprints with polyBERT fingerprints. polyBERT may also be used to directly design polymers based on fingerprints (that can be related to properties) using polyBERT's decoder that has been trained during the self-supervised learning. This, however, requires retraining and structural updates to polyBERT and is thus part of a future work.

## Methods

### PSMILES canonicalization

The string representations of homopolymer repeat units in this work are PSMILES strings. PSMILES strings follow the SMILES[24] syntax definition but use two stars to indicate the two endpoints of the polymer repeat unit (e.g., `[*]CC[*]` for polyethylene). The raw PSMILES syntax is non-unique; i.e., the same polymer may be represented using many PSMILES strings; canonicalization is a scheme to reduce the different PSMILES strings of the same polymer to a singel unique canonicalized PSMILES string. polyBERT requires canonicalized PSMILES strings

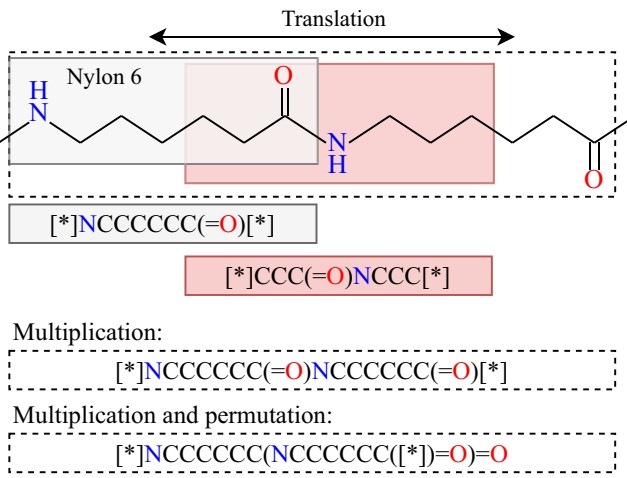

**Fig. 6 | Translational, multiplicative, and permutational variances of Polymer Simplified Molecular-Input Line-Entry System (PSMILES) strings.** The gray and red boxes represent the smallest repeat unit of poly(hexano-6-lactam) (Nylon 6). The red box can be translated to match the black box. The dashed boxes show the second smallest repeat unit (two-fold repeat unit) of Nylon 6.

because polyBERT fingerprints change with different writings of PSMILES strings. In contrast, PG fingerprints are invariant to the way of writing PSMILES strings and, thus, do not require canonicalization. Figure 6 shows three variances of PSMILES strings that leave the polymer unchanged. The translational variance of PSMILES strings allows to move the repeat unit window of polymers (cf., white and red box). The multiplicative variance permits to write polymers as multiples of the repeat unit (e.g., twofold repeat unit of Nylon 6), while the permutational variance stems from the SMILES syntax definition[24] and allows syntactical permutations of PSMILES strings that leave the polymer unchanged.

For this work, we developed the `canonicalize_psmiles` Python package that finds the canonical form of PSMILES strings in four steps; (i) it finds the shortest PSMILES string by searching and removing repetition patterns, (ii) it connects the polymer endpoints to create a periodic PSMILES string, (iii) it canonicalizes the periodic PSMILES string using RDKit's[41] canonicalization routines, (iv) it breaks the periodic PSMILES string to create the canonical PSMILES string. The `canonicalize_psmiles` package is available at https://github.com/Ramprasad-Group/canonicalize_psmiles.

### Polymer fingerprinting

Fingerprinting converts geometric and chemical information of polymers (based upon the PSMILES string) to machine-readable numerical representations in the form of vectors. These vectors are the polymer fingerprints and can be used for property predictions, similarity searches, or other tasks that require numerical representations of polymers.

We compare the polyBERT fingerprints, developed in this work, with the handcrafted Polymer Genome (PG) polymer fingerprints. PG fingerprints capture key features of polymers at three hierarchical length scales[8,20]. At the atomic scale (1st level), PG fingerprints track the occurrence of a fixed set of atomic fragments (or motifs)[21]. The block scale (2nd level) uses the Quantitative Structure-Property Relationship (QSPR) fingerprints[18,44] for capturing features on larger length-scales as implemented in the cheminformatics toolkit RDKit[41]. The chain scale (3rd level) fingerprint components deal with "morphological descriptors" such as the ring distance or length of the largest side-chain[44]. The PG fingerprints are developed within the Ramprasad research group and used, for example, at https://PolymerGenome.org. More details can be found in References[8,44].

As discussed recently[6,11], we sum the composition-weighted polymer fingerprints to compute copolymer fingerprints $\mathcal{F} = \sum_i^N \mathbf{F}_i c_i$, where $N$ is the number of comonomers in the copolymer, $\mathbf{F}_i$ the $i^{\text{th}}$ comonomer fingerprint vector, and $c_i$ the fraction of the $i^{\text{th}}$ comonomer. This approach renders copolymer fingerprints invariant to the order in which one may sort the comonomers and satisfies the two main demands of uniqueness and invariance to different (but equivalent) periodic unit specifications. While the current finger-printing scheme is most appropriate for random copolymers, other copolymer flavors may be encoded by adding additional fingerprint components. Contrary to homopolymer fingerprints, copolymer fingerprints may not be interpretable (e.g., the composition-weighted sum of the fingerprint component "length of largest side-chain" of two homopolymers has no physical meaning).

### Multitask neural networks

Multitask deep neural networks simultaneously learn multiple polymer properties to utilize inherent correlations of properties in data sets. The training protocol of the concatenation-conditioned multi-task predictors follows state-of-the-art techniques involving five-fold cross-validation and a meta learner that forecasts the final property values based upon the ensemble of cross-validation predictors[6,7,11]. Supplementary Figure S2 details this process. After shuffling, we split the data set into two parts and use 80% for the five cross-validation models and for validating the meta learners. 20% of the data set is used for training the meta learners. We use the Hyperband method[58] of the Python package KerasTuner[59] to fully optimize all hyperparamters of the neural networks, including the number of layers, number of nodes, dropout rates, and activation functions. The Hyperband method finds the best set of hyperparameters by minimizing the Mean Squared Error (MSE) loss function. We perform data set stratification of all splits based on the polymer properties. The multitask deep neural networks are implemented using the Python API of TensorFlow[60].

### CO$_2$ emission and timing

Experiments were conducted using a private infrastructure, which has an estimated carbon efficiency of 0.432 kgCO$_2$eq kWh$^{-1}$. A total of 31 h of computations were performed on four Quadro-GP100-16GB (thermal design power of 235 W) for training polyBERT. Total emissions are estimated to be 12.6 kgCO$_2$eq. About 8 h of computations on four GPUs were necessary for training the cross-validation and meta learner models with an estimated emission of 3.3 kgCO$_2$eq for polyBERT and Polymer Genome fingerprints, respectively. The total emissions for predicting 29 properties for 100 million hypothetical polymers are estimated to be 5.5 kgCO$_2$eq, taking a total of 13.5 h. Estimations were conducted using the Machine Learning Impact calculator presented in ref. 61.

### Reporting summary

Further information on research design is available in the Nature Portfolio Reporting Summary linked to this article.

## Data availability

The data set of 100 million hypothetical polymers with 29 predicted properties is available for academic use at https://doi.org/10.5281/zenodo.7766806.

## Code availability

The polyBERT code is available for academic use at https://github.com/Ramprasad-Group/polyBERT and Zenodo[62]. The trained poly-BERT model is available at https://huggingface.co/kuelumbus/polyBERT. The Python package for canonicalizing PSMILES strings is available at https://github.com/Ramprasad-Group/canonicalize_psmiles. polyBERT-based property predictions will be made

accessible through the polymer informatics platform Polymer Genome at https://PolymerGenome.org.

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

## Acknowledgements

C.K. thanks the Alexander von Humboldt Foundation for financial support. We acknowledge funding from the Office of Naval Research through a Multidisciplinary University Research Initiative grant (N00014-17-1-2656) and the National Science Foundation (#1941029).

## Author contributions

C. K. designed, trained and evaluated the machine learning models and drafted this paper. The work was conceived and guided by R. R. All authors discussed results and commented on the manuscript.

## Competing interests

R.R. is the founder of the company Matmerize, Inc., that intends to provide polymer informatics services. A provisional patent has been filed by the Georgia Tech Research Corporation, Atlanta, GA; Inventors: Rampi Ramprasad, and Christopher Kuenneth; Application Number: 63/374,761; Status: patent pending; Aspect covered in the patent application: transformer based informatics pipeline for polymer representation and property predictions.

## Additional information

**Peer review information** : *Nature Communications* thanks the anonymous, reviewer(s) for their contribution to the peer review of this work. A peer review file is available.

