## [Peer Review File · Nature Communications]

polyBERT: A chemical language model to enable fully machine-driven ultrafast polymer informaticsREVIEWER COMMENTS

Reviewer #1 (Remarks to the Author):

The most important tool in machine learning-driven property prediction is a material descriptor that adequately represents the compositional, structural, and physicochemical features of any given material. This paper proposes to use a machine-learned distributed representation of the fundamental language model polyBERT that was trained on SMILES strings of approximately 100 million synthesized and virtually created polymers to describe the chemical structure of polymer repeating units. The training dataset, Python source code for model training, and pre-trained models are available at open repositories. The research topic is timely, thus receiving much attention from data-driven materials research. The manuscript is generally concise and clearly written. I consider the manuscript publishable if the following minor comments are addressed:

1. The source of the 13766 polymers synthesized so far should be listed.
2. The number and distribution of tokens identified by the SentencePiece tokenizer should be reported with a more comprehensive set of their examples.
3. Compared to homopolymers, the data set of copolymers is quite small. Therefore, the representation ability for copolymers may be inferior to that of homopolymers. I request that you separately compare the prediction performance of homopolymers and copolymers in benchmarking the property prediction tasks.
4. I understood that a copolymer is given into the model simply by concatenating the SMILES strings of its monomers with periods. If this understanding is correct, the model will not be able to distinguish random, alternating copolymers, etc. This should be remarked as a limitation.
5. It is obvious that the computation times of polyBERT and PG are linear with respect to the number of polymers, since the computation of multiple polymers is completely independent. In the last paragraph on page 12, there is no need to emphasize that it is linear.
6. In the manuscript, it is stated that the accuracy of the property prediction was evaluated on the validation set, but is this a mistake for the test set? The validation set should never be used to evaluate the prediction performance of a model.

7. The difference between ordinary fingerprints such as PG and the polyBERT representations should be investigated in more detail. In the property prediction benchmark, the reported prediction accuracy of the two is almost equal. One question here is; does using a descriptor that concatenates PG and polyBERT fingerprints improve prediction accuracy? If there are no intrinsic differences between the two, the prediction accuracy would not be different from models with PG and polyBERT fingerprints alone.

The difference between the two can also be examined as follows: For a set of test polymers, their similarity matrix can be computed with the PG or polyBERT features, respectively. These two matrices can be compared visually (e.g. using heatmaps) or quantitatively to identify a subset of polymers differently represented by the two descriptors.

Reviewer #2 (Remarks to the Author):

The manuscript by C. Kuenneth et al. presents a very interesting framework built on a polymer chemical language model to drive the discovery of novel polymers with targeted properties. Transformer models are currently revolutionizing the field of materials informatics and this work is the latest example of it. The article is well written and well motivated, and the results are meaningful. I believe the polyBERT framework will be enormously useful to the community. I recommend the manuscript for publication in Nature Communications. I have the following minor comments for the authors to address before publication:

1. I am curious about the generation of the 100 million hypothetical PSMILES strings to train the polyBERT model. Is 100 million just an arbitrarily big number necessary for training, or can this purpose be achieved with a smaller dataset?
2. In table 1, it appears there is a heavy mismatch between datasets for different properties, as well as some mixing and matching going on in terms of experimental and DFT datasets. From Figure 5, it appears that every property is predicted highly accurately. Please shed some light on this- how much does the size and source of data affect the actual property predictions?
3. There are a couple of spelling errors, such as the word "originallydro" on page 18. Please check.

Reviewer #3 (Remarks to the Author):

In this work, Kuenneth and Ramprasad propose polyBERT, a deep learning language model for polymer fingerprint and multitask property predictions. In particular, each polymer is converted to polymer SMILES (PSMILES) and is fed into polyBERT, which is built upon DeBERTa. The model is first pretrained on 100M hypothetical PSMILES data through masked language modeling. The pretrained model is transferred to multitask property prediction tasks. polyBERT demonstrates rival prediction accuracy in comparison to handcrafted Polymer Genome fingerprint under multitask learning. While polyBERT is much faster than Polymer Genome especially when leveraging GPU hardware. Investigations via UMAP and attention map show polyBERT learns information of polymers through self-supervised pretraining. This work unveils the potential of using large language model (i.e., Transformer) for polymer informatics and can be an interesting contribution to a broad community. However, I have the following concerns before the paper being published on Nature Communications.

1. Compared with polymer Genome (PG), polyBERT shows rival or slightly worse prediction accuracy in most benchmarks. Thus, the major benefit of polyBERT compared with polymer Genome (PG) is the prediction speed. As shown in Figure 4, when dealing with 10^4 PSMILES, PG takes 4-5 hours while polyBERT takes only few seconds when using GPU, which is a huge improvement. However, the computational time using PG to deal with 10^4 PSMILES is not totally unacceptable. The authors are suggested to comment on the magnitude of polymer data for virtual screening in practice to further validate the benefits of polyBERT. Also, in terms of the computational time, what is the pretraining plus finetuning time of polyBERT when comparing to PG. I suggest the authors include training time for fair comparison of the computation time.
2. In page 5, the authors mention “We also note that unlike Transformers graph neural networks are usually trained end-to-end, i.e., their latent space representations (fingerprints) are learned under supervision with polymer properties.” It is true that pretrained Transformer has been a widely used scheme in machine learning. However, there are also many works investigating pretraining graph neural networks, e.g., <https://openreview.net/forum?id=HJIWWJSFDH>, <https://www.nature.com/articles/s42256-022-00447-x>, <https://www.nature.com/articles/s42256-021-00438-4>, <https://proceedings.neurips.cc/paper/2021/hash/85267d349a5e647ff0a9edcb5ffd1e02-Abstract.html>. I suggest the authors include necessary discussions on GNN pretraining as well.
3. polyBERT is built upon DeBERTa, a variant of Transformer-based model. I suggest the authors provide more discussion about why DeBERTa, instead of other BERT-like models, is preferred for polymers in this work.
4. Using BRICS decomposition can generate hypothetical PSMILES. However, it can also generate unrealistic polymers. Will this affect the performance of the model?
5. This work introduces PSMILES canonicalization, which helps get clean polymer representations. Can un-canonicalized PSMILES used for augmentation of training data?
6. PSMILES used in this work is based on SMILES with stars to indicate repeat units. Have the authors consider using existing sequence representations like BigSMILES (<https://pubs.acs.org/doi/10.1021/acscentsci.9b00476>)?

7. In multitask learning shown in FigureS2, selector vectors are determined based on properties, e.g., electrical, mechanical, permeability, etc. Will multitask learning among different properties help? Did the authors have data analysis on the correlation different polymer properties?

8. How does polyBERT scale with the number of pretraining data. Namely, does the performance of polyBERT increase as the number of pretraining data increases and as the size of the model increases?

9. The authors include the predicted property distributions of 100M hypothetical polymers via proposed polyBERT. Are there polymers with desired predicted properties that provide insights on polymer design? The authors are recommended to show some cases to further validate the potential of polyBERT in practice.

10. Page 8 writes "The validation F1-score is > 99." I suppose this is a typo and the authors mean that F1-score is > 0.99.

REVIEWER COMMENTS

Our general response: We thank the reviewers for their thorough evaluation of our manuscript and their insightful remarks. Our detailed reply to each of the reviewers' comments, along with the corresponding modifications made to the manuscript, are outlined below.

Reviewer #1 (Remarks to the Author):

The most important tool in machine learning-driven property prediction is a material descriptor that adequately represents the compositional, structural, and physicochemical features of any given material. This paper proposes to use a machine-learned distributed representation of the fundamental language model polyBERT that was trained on SMILES strings of approximately 100 million synthesized and virtually created polymers to describe the chemical structure of polymer repeating units. The training dataset, Python source code for model training, and pre-trained models are available at open repositories. The research topic is timely, thus receiving much attention from data-driven materials research. The manuscript is generally concise and clearly written. I consider the manuscript publishable if the following minor comments are addressed:

1. The source of the 13766 polymers synthesized so far should be listed.

Response: These 13766 polymers are the polymers contained in the data set outlined in Table 1. These have been previously discussed in multiple publications [1-10]. We have added a statement to indicate this in the manuscript:

“We use the Breaking Retrosynthetically Interesting Chemical Substructures (BRICS)¹⁹ method (as implemented in RDKit⁴¹) to decompose previously synthesized 13766 polymers (**all monomers of the data set outline in Table 1, see below**) into 4424 unique chemical fragments”

[1] Kuenneth, C.; Schertzer, W.; Ramprasad, R. Copolymer Informatics with Multitask Deep Neural Networks. *Macromolecules* 2021, 54, 5957–5961, DOI: 10.1021/acs.macromol.1c00728.

[2] Kuenneth, C.; Rajan, A. C.; Tran, H.; Chen, L.; Kim, C.; Ramprasad, R. Polymer informatics with multi-task learning. *Patterns* 2021, 2, 100238, DOI: 10.1016/j.patter.2021.100238.

[3] Kuenneth, C.; Lalonde, J.; Marrone, B. L.; Iverson, C. N.; Ramprasad, R.; Pilania, G. Bioplastic design using multitask deep neural networks. *Communications Materials* 2022, 3, 96, DOI: 10.1038/s43246-022-00319-2.

[4] Jha, A.; Chandrasekaran, A.; Kim, C.; Ramprasad, R. Impact of dataset uncertainties on machine learning model predictions: the example of polymer glass transition temperatures. *Modelling and Simulation in Materials Science and Engineering* 2019, 27, 024002, DOI: 10.1088/1361-651X/aaf8ca.

- [5] Kim, C.; Chandrasekaran, A.; Jha, A.; Ramprasad, R. Active-learning and materials design: the example of high glass transition temperature polymers. *MRS Communications* 2019, 9, 860–866, DOI: 10.1557/mrc.2019.78.
- [6] Kim, C.; Chandrasekaran, A.; Huan, T. D.; Das, D.; Ramprasad, R. Polymer Genome: A Data-Powered Polymer Informatics Platform for Property Predictions. *The Journal of Physical Chemistry C* 2018, 122, 17575–17585, DOI: 10.1021/acs.jpcc.8b02913.
- [7] Patra, A.; Batra, R.; Chandrasekaran, A.; Kim, C.; Huan, T. D.; Ramprasad, R. A multi-fidelity information-fusion approach to machine learn and predict polymer bandgap. *Computational Materials Science* 2020, 172, 109286, DOI:10.1016/j.commatsci.2019.109286.
- [8] Chen, L.; Kim, C.; Batra, R.; Lightstone, J. P.; Wu, C.; Li, Z.; Deshmukh, A. A.; Wang, Y.; Tran, H. D.; Vashishta, P.; Sotzing, G. A.; Cao, Y.; Ramprasad, R. Frequency-dependent dielectric constant prediction of polymers using machine learning. *npj Computational Materials* 2020, 6, 61, DOI: 10.1038/s41524-020-0333-6.
- [9] Venkatram, S.; Kim, C.; Chandrasekaran, A.; Ramprasad, R. Critical Assessment of the Hildebrand and Hansen Solubility Parameters for Polymers. *Journal of Chemical Information and Modeling* 2019, 59, 4188–4194, DOI: 10.1021/acs.jcim.9b00656.
- [10] Zhu, G.; Kim, C.; Chandrasekaran, A.; Everett, J. D.; Ramprasad, R.; Lively, R. P. Polymer genome-based prediction of gas permeabilities in polymers. *Journal of Polymer Engineering* 2020, 40, 451–457, DOI: 10.1515/polyeng-2019-0329

2. The number and distribution of tokens identified by the SentencePiece tokenizer should be reported with a more comprehensive set of their examples.

Response: Thank you for this comment. When working on this comment, we identified an inaccuracy in our manuscript that states that tokens have been determined using a pretraining step. Although this is true, we have found that the specified limit of the number of tokens matches the number of predefined tokens. This means that although we trained the tokenizer on the data set, no additional tokens (except auxiliary tokens) were added to the token library. We have made the following changes to the “polyBERT” subsection in the manuscript to clarify this:

“Second, polyBERT tokenizes canonical PSMILES strings using the SentencePiece⁵³ tokenizer and a total of 265 tokens. The tokens include common PSMILES characters such as the uppercased and lowercased 118 elements of the periodic table of elements, numbers ranging from 0 to 9, and special characters like [*], (,), =, among others. This ensures that the tokenizer covers the entire PSMILES strings vocabulary and is a similar approach to that in Ref. [1]. A full token list can be found at the Github repository (see the Data and Code Availability section).”

[1] Ahmad, W.; Simon, E.; Chithrananda, S.; Grand, G.; Ramsundar, B. ChemBERTa-2: Towards Chemical Foundation Models. 2022, DOI: 10.48550/arXiv.2209.01712.

3. Compared to homopolymers, the data set of copolymers is quite small. Therefore, the representation ability for copolymers may be inferior to that of homopolymers. I request that you

separately compare the prediction performance of homopolymers and copolymers in benchmarking the property prediction tasks.

Response: To address this comment, we have added Table S2 and S3 to the SI that show the RMSE and R^2 values separately for homopolymers and copolymers. As expected, and also found in Ref [1], the predictions for copolymers are better in the case of thermal and mechanical properties but slightly worse for the gas permeabilities.

We have added the following statement to the “Property Prediction” subsection:

“Root-mean-square error (RMSE) and R^2 values for the cross-validation and meta learner models can be found in Supplementary Table S1-S3 for all polymers, homopolymers, and copolymers, respectively. We find the prediction accuracy to be better for thermal and mechanical properties of copolymers (relative to that for homopolymers) and slightly worse for the gas permeabilities, similar to previous findings [1].”

[1] Kuenneth, C.; Schertzer, W.; Ramprasad, R. Copolymer Informatics with Multitask Deep Neural Networks. *Macromolecules* 2021, 54, 5957–5961, DOI: 10.1021/acs.macromol.1c00728

4. I understood that a copolymer is given into the model simply by concatenating the SMILES strings of its monomers with periods. If this understanding is correct, the model will not be able to distinguish random, alternating copolymers, etc. This should be remarked as a limitation.

Response: The copolymer fingerprints are generated by summing the composition-weighted homopolymer fingerprints, as explained in the 3rd paragraph of the "Polymer Fingerprinting" subsection of the "Methods" section. The reviewer is correct in that this method is currently unable to differentiate between copolymer types, such as gradient or block copolymers. However, our fingerprinting method can easily be extended to deal with different copolymer types by, for example, adding an additional fingerprint component that encodes the copolymer type. Moreover, currently, our data set contains no information on the different copolymer types, and so we have adopted the current fingerprinting method.

This is a limitation and now discussed in the “Data Set” subsection:

“Our copolymer data points are for random copolymers, which are adequately handled by our adopted fingerprinting strategy (see Methods section). Alternating copolymers are treated as homopolymers with appropriately defined repeat units for fingerprinting purposes. Other flavors of copolymers may also be encoded by adding additional fingerprint components.”

An additional statement is also added to the “Polymer Fingerprinting” subsection of the “Methods” section:

“While the current fingerprinting scheme is most appropriate for random copolymers, other copolymer flavors may be encoded by adding additional fingerprint components.”

5. It is obvious that the computation times of polyBERT and PG are linear with respect to the number of polymers, since the computation of multiple polymers is completely independent. In the last paragraph on page 12, there is no need to emphasize that it is linear.

Response: Even though obvious to experts, a linear dependence of the computation times of polyBERT (CPU), polyBERT (GPU), and Polymer Genome fingerprints might not be so clear to the general audience. However, after reviewing the reviewer's suggestion, we have de-emphasized this point as follows.

The statement now reads:

“Not surprisingly, the computations of polyBERT and PG fingerprints scale nearly linearly with the number of PSMILES strings although their performance (**i.e., pre-factor**) can be quite different, as shown in the log-log scaled Figure 4.”

6. In the manuscript, it is stated that the accuracy of the property prediction was evaluated on the validation set, but is this a mistake for the test set? The validation set should never be used to evaluate the prediction performance of a model.

Response: Indeed, the cross-validation performance values in Figure 5a are computed using the validation data sets. This is a fair procedure as both the polyBERT and Polymer Genome fingerprint-based models are tested using the exact same data points (identical data set splits). An additional data set, which is completely separate from the cross-validation process, is kept to train the meta learner.

7. The difference between ordinary fingerprints such as PG and the polyBERT representations should be investigated in more detail. In the property prediction benchmark, the reported prediction accuracy of the two is almost equal. One question here is; does using a descriptor that concatenates PG and polyBERT fingerprints improve prediction accuracy? If there are no intrinsic differences between the two, the prediction accuracy would not be different from models with PG and polyBERT fingerprints alone.

The difference between the two can also be examined as follows: For a set of test polymers, their similarity matrix can be computed with the PG or polyBERT features, respectively. These two matrices can be compared visually (e.g. using heatmaps) or quantitatively to identify a subset of polymers differently represented by the two descriptors.

Response: There are differences between the PG and polyBERT fingerprints. In the beginning of the “Fingerprints” section, we pointed out numeric differences, such as the number of fingerprint components and densities. Moreover, PG fingerprints are chemically relevant polymer representations and, for example, count chemical groups present in the polymer, while polyBERT fingerprints are dense representations learned from a large data set through attention and feed-forward neural networks (Transformers).

Although an interesting thought, we believe that concatenating polyBERT and PG fingerprints would undermine the large speed advantages of polyBERT and is not a practicable path forward. Moreover, it would more than double the dimensionality of the fingerprints, reducing training and inference speed.

Because the fingerprints are intrinsically different and have no 1-to-1 mapping (as pointed out above and in the manuscript in the “Fingerprints” section), the pure fingerprint components cannot be compared side-by-side or in a heatmap. However, we show UMAP plots for polyBERT and PG fingerprints in Figure 2. These UMAP plots visualize a 2-dimensional representation of the fingerprint spaces that allow to compare the fingerprints side-by-side.

Reviewer #2 (Remarks to the Author):

The manuscript by C. Kuenneth et al. presents a very interesting framework built on a polymer chemical language model to drive the discovery of novel polymers with targeted properties. Transformer models are currently revolutionizing the field of materials informatics and this work is the latest example of it. The article is well written and well motivated, and the results are meaningful. I believe the polyBERT framework will be enormously useful to the community. I recommend the manuscript for publication in Nature Communications. I have the following minor comments for the authors to address before publication:

1. I am curious about the generation of the 100 million hypothetical PSMILES strings to train the polyBERT model. Is 100 million just an arbitrarily big number necessary for training, or can this purpose be achieved with a smaller dataset?

Response: Through training polyBERT with 100 million hypothetical PSMILES strings, we intend to produce meaningful machine learning models that can make reliable predictions in the entire polymer universe. However, we understand that our approach of producing the 100 million hypothetical polymers through BRICS and a database of 13766 synthesized polymers may have intrinsic limitations that do not allow us to produce polymers in all facets of the polymer universe.

Transformer models based on BERT or similar architectures are typically trained on large corpora comprising several billion words. In chemistry, the training sizes of corpora continue to rise, with ChemBERTa utilizing 10 million, SMILES-BERT using 18.7 million, and ChemBERTa-2 using 77 million SMILES strings. To augment the dataset size (although for polymers) even further, polyBERT uses 100 million PSMILES strings for training. Here, we wanted to ensure that our data set covers a large enough part of the chemical space of synthesized and synthesizable polymers.

To reflect this in the text, we have added the following statement in the “polyBERT” subsection:

“Using 100 million PSMILES strings is the latest example of training a chemistry-related language model with a large data set and follows the trend of growing data sets in this

discipline, with ChemBERTa using 10 million, SMILES-BERT using 18.7 million, and ChemBERTa-2 using 77 million SMILES strings. [1]"

[1] Ahmad, W.; Simon, E.; Chithrananda, S.; Grand, G.; Ramsundar, B. ChemBERTa-2: Towards Chemical Foundation Models. 2022, DOI: 10.48550/arXiv.2209.01712.

2. In table 1, it appears there is a heavy mismatch between datasets for different properties, as well as some mixing and matching going on in terms of experimental and DFT datasets. From Figure 5, it appears that every property is predicted highly accurately. Please shed some light on this- how much does the size and source of data affect the actual property predictions?

Response: The different quantities and sources of the polymer properties in our data set come from past data collection and in-house data generation efforts. Our multitask learning strategy is capable of learning from differently generated and sized data sets and, at the same time, picking up inherent correlations between data points (and property data sets) to improve its overall prediction performance. Because our property data sets have been collected and refined over many years, a high overall prediction performance is expected for the properties.

However, we also find properties like the crystallization tendencies (X_C and X_e), cohesive energy density (δ), limiting oxygen index (O_i), or heat capacity (c_p) whose performance values are not so good ($R^2 \leq 0.61$). This may indicate that these data sets are less curated, too noisy (i.e., prone to measurement uncertainties), too small, or hard to learn using our current machine learning techniques. We also note that although the general prediction performance is important, the performance difference of polyBERT and PG-based models is of main interest in this study.

3. There are a couple of spelling errors, such as the word "originallydro" on page 18. Please check.

Response: Thank you for pointing this out. We have corrected this spelling error and carefully checked and corrected the manuscript for other typos.

Reviewer #3 (Remarks to the Author):

In this work, Kuenneth and Ramprasad propose polyBERT, a deep learning language model for polymer fingerprint and multitask property predictions. In particular, each polymer is converted to polymer SMILES (PSMILES) and is fed into polyBERT, which is built upon DeBERTa. The model is first pretrained on 100M hypothetical PSMILES data through masked language modeling. The pretrained model is transferred to multitask property prediction tasks. polyBERT demonstrates rival prediction accuracy in comparison to handcrafted Polymer Genome fingerprint under multitask learning. While polyBERT is much faster than Polymer Genome especially when leveraging GPU hardware. Investigations via UMAP and attention map show polyBERT learns information of polymers through self-supervised pretraining. This work unveils the potential of using large language model (i.e., Transformer) for polymer informatics and can

be an interesting contribution to a broad community. However, I have the following concerns before the paper being published on Nature Communications.

1. Compared with polymer Genome (PG), polyBERT shows rival or slightly worse prediction accuracy in most benchmarks. Thus, the major benefit of polyBERT compared with polymer Genome (PG) is the prediction speed. As shown in Figure 4, when dealing with 10^4 PSMILES, PG takes 4-5 hours while polyBERT takes only few seconds when using GPU, which is a huge improvement. However, the computational time using PG to deal with 10^4 PSMILES is not totally unacceptable. The authors are suggested to comment on the magnitude of polymer data for virtual screening in practice to further validate the benefits of polyBERT. Also, in terms of the computational time, what is the pretraining plus finetuning time of polyBERT when comparing to PG. I suggest the authors include training time for fair comparison of the computation time.

Response: Thank you for this comment. The space of polymers is as large as we would like it to be! And this places demands on the prediction speed which determines how much of the space can be explored. To address the size of the polymer universe, we have added the following text in the 2nd paragraph of the “Discussion” section:

“The total polymer universe is gigantic, but currently limited by experimentation, manufacturing techniques, resources, and economical aspects. Contemplating different polymer types such as homopolymers, copolymer, and polymer blends, novel undiscovered polymer chemistries, additives, and processing conditions, the number of possible polymers in the polymer universe is truly limitless. Searching this extraordinarily large space enabled by property predictions is limited by the prediction speed. The accurate prediction of 29 properties for 100 million hypothetical polymers in a reasonable time demonstrates that polyBERT is an enabler to extensive explorations of this gigantic polymer universe at scale.”

Regarding the timing of polyBERT, the training time of polyBERT is mentioned in the “CO₂ Emission” section (31h on 4 GPUs), which we renamed to “CO₂ Emission and Timing”. polyBERT is not fine-tuned on a specific downstream task but directly used for fingerprint predictions after training on 100 million hypothetical PSMILES strings. Relevant training and prediction times include:

- 31h for training polyBERT on 100 million PSMILES strings
- 8h for training the cross-validation and meta learner models for PG and polyBERT fingerprints, respectively
- 13.5h for predicting all properties of the 100m PSMILES strings
- Computation times of PG and polyBERT fingerprints with respect to the number of PSMILES strings are shown in Figure 4. This plot is a fair comparison of the polyBERT and PG prediction pipelines because training polyBERT is a one-time task, and the time for property prediction is similar for PG and polyBERT fingerprints.

The revised “CO₂ Emission and Timing” section reads:

CO₂ Emission and Timing

“Experiments were conducted using a private infrastructure, which has an estimated carbon efficiency of 0.432 kgCO₂eq kWh⁻¹. A total of 31 hours of computations were performed on four Quadro-GP100-16GB (thermal design power of 235 W) for training polyBERT. Total emissions are estimated to be 12.6 kgCO₂eq. About 8 hours of computations on four GPUs were necessary for training the cross-validation and meta learner models with an estimated emission of 3.3 kgCO₂eq for polyBERT and Polymer Genome fingerprints, respectively. The total emissions for predicting 29 properties for 100 million hypothetical polymers are estimated to be 5.5 kgCO₂eq, taking a total of 13.5 hours. Estimations were conducted using the Machine Learning Impact calculator presented in Reference 60.”

2. In page 5, the authors mention “We also note that unlike Transformers graph neural networks are usually trained end-to-end, i.e., their latent space representations (fingerprints) are learned under supervision with polymer properties.” It is true that pretrained Transformer has been a widely used scheme in machine learning. However, there are also many works investigating pretraining graph neural networks, e.g., <https://openreview.net/forum?id=HJIWWJSFDH>, <https://www.nature.com/articles/s42256-022-00447-x>, <https://www.nature.com/articles/s42256-021-00438-4>, <https://proceedings.neurips.cc/paper/2021/hash/85267d349a5e647ff0a9edcb5ffd1e02-Abstract.html>. I suggest the authors include necessary discussions on GNN pretraining as well.

Response: Thank you for pointing this out. We have included the citations and a short discussion in the Introduction section:

“We note that self-supervised graph neural networks^{38,39} have recently been developed that learn the molecule graph through atom, bond, and subgraph masking, an approach similar to Transformers.”

3. polyBERT is built upon DeBERTa, a variant of Transformer-based model. I suggest the authors provide more discussion about why DeBERTa, instead of other BERT-like models, is preferred for polymers in this work.

Response: During the development of polyBERT, we have tested BERT, RoBERTa, and DistilBERT as the basis of polyBERT. DeBERTa turned out to perform best. We have added the following statement to the text:

“We chose the DeBERTa model as the foundation of polyBERT because it outperformed other BERT-like models (BERT²⁹, RoBERTa³³, and DistilBERT⁵²) in our preliminary tests.”

4. Using BRICS decomposition can generate hypothetical PSMILES. However, it can also generate unrealistic polymers. Will this affect the performance of the model?

Response: The BRICS decomposition and reconstruction produce hypothetical polymers that have not been synthesized before (however, chemically nonsensical materials will not be produced). Certainly, some of them may not be synthesizable using current methods but future

techniques may allow their synthesis. polyBERT's training and performance should not be affected by these polymers because they are used for learning a numerical representation (fingerprints) of polymers only. On the contrary, we believe that including such "not-yet-been-synthesized" polymers extends the prediction capabilities of polyBERT to "new-to-the-world" polymers in the polymer universe making polyBERT robust for future tasks. We re-emphasize that all 100 million hypothetical polymers in our data set are chemically valid polymers.

5. This work introduces PSMILES canonicalization, which helps get clean polymer representations. Can un-canonicalized PSMILES used for augmentation of training data?

Response: This is an interesting question. It is important to canonicalize PSMILES strings for the generation of polymer fingerprints because polyBERT may produce different fingerprints for a set of non-canonicalized, but chemically identical, PSMILES strings. Consequently, the predicted properties of a set of chemically identical, but non-canonicalized, PSMILES strings would be different. This is the main motivation for developing a canonicalization routine for PSMILES.

For training polyBERT, we believe that including randomized (non-canonicalized) PSMILES strings to the data set would not alter the prediction performance for polymers because these new un-canonicalized learned spaces would never be used by polyBERT. However, when used for molecules (as discussed in the Section "Other Advantages of polyBERT: Beyond Speed and Accuracy"), training using such an augmented data set might be beneficial.

6. PSMILES used in this work is based on SMILES with stars to indicate repeat units. Have the authors consider using existing sequence representations like BigSMILES (<https://pubs.acs.org/doi/10.1021/acscentsci.9b00476>)?

Response: BigSMILES strings are useful for describing the stochastic nature of polymers using a chemical language. Our current data set includes random copolymers only and does not benefit from using BigSMILES. However, when polymer data with full stochastic information become available, the BigSMILES representation might be a good fit.

7. In multitask learning shown in FigureS2, selector vectors are determined based on properties, e.g., electrical, mechanical, permeability, etc. Will multitask learning among different properties help? Did the authors have data analysis on the correlation different polymer properties?

Response: This is another interesting question. Indeed, we have considered a single combined multitask model and separate multitask models for property categories in a previous study [1]. We found that properties in the currently used categories have strong correlations and train well together, producing high-performance models. A correlation plot of polymer properties can be found in Figure 2 of Ref [1].

To address the reviewers question, we have added the following statement to the text:

“In our previous study⁷, we observed that these property categories resulted in the development of models exhibiting superior performance.”

[1] Kuenneth, C.; Rajan, A. C.; Tran, H.; Chen, L.; Kim, C.; Ramprasad, R. Polymer informatics with multi-task learning. *Patterns* 2021, 2, 100238, DOI: 10.1016/j.patter.2021.100238

8. How does polyBERT scale with the number of pretraining data. Namely, does the performance of polyBERT increase as the number of pretraining data increases and as the size of the model increases?

Response: In the very beginning of developing polyBERT, we started with a much smaller data set and found that increasing the data set improves the learning accuracy of polyBERT. Similarly, we observed that increasing the model parameters (number of hidden layers and attention heads) improves the prediction accuracy but worsens prediction times.

Please also see the related question 1 of reviewer #2.

9. The authors include the predicted property distributions of 100M hypothetical polymers via proposed polyBERT. Are there polymers with desired predicted properties that provide insights on polymer design? The authors are recommended to show some cases to further validate the potential of polyBERT in practice.

Response: The data set of 100 million hypothetical polymers with 29 predicted properties showcases polyBERT’s potential for exploring the polymer universe. We use this large data set to identify ranges of values that can be achieved for the properties considered. The ranges are shown via distribution plots in Figure S15-S20 that highlight the limits of properties potentially achievable using this data set. Moreover, we are confident that the data set contains additional interesting information and correlations that can be investigated, although such explorations are beyond the scope of this study. These aspects are highlighted by the following added text in Section “Property Prediction”:

“In addition, a data set of this magnitude presents numerous opportunities for obtaining fascinating insights and practical applications. For example, it can be utilized in future studies to establish standardized benchmarks for testing and evaluating ML models in the domain of polymer informatics. The data set may also reveal structure-property information that provides guidance for design rules, helps to identify unexplored areas to search for new polymers, or facilitates direct selection of polymers with specific properties through nearest neighbor searches, as evidenced in a recent study [1]. A possible future evolution of the data set may also contain subspaces of distinct polymer classes, such as biodegradable or low-carbon polymer classes. However, these aspects are beyond the scope of this study.”

[1] Kuenneth, C.; Lalonde, J.; Marrone, B. L.; Iverson, C. N.; Ramprasad, R.; Pilia, G. Bioplastic design using multitask deep neural networks. *Communications Materials* 2022, 3, 96, DOI: 10.1038/s43246-022-00319-2

10. Page 8 writes “The validation F1-score is > 99.” I suppose this is a typo and the authors mean that F1-score is > 0.99.

Response: Thanks for finding this typo. We have corrected this in the manuscript.

REVIEWERS' COMMENTS

Reviewer #1 (Remarks to the Author):

The authors have addressed almost all comments; I recommend the manuscript for publication.

Reviewer #2 (Remarks to the Author):

I thank the authors for addressing all reviewer comments. I am happy to accept the manuscript for publication.

Reviewer #3 (Remarks to the Author):

I appreciate the authors' efforts in the rebuttal, which have substantially improved the quality of the manuscript. I accept the responses from the authors and recommend the publication of this paper.